# CLASSLESS ASSOCIATION USING NEURAL NETWORKS

**Federico Raue[1,2], Sebastian Palacio[2], Andreas Dengel[1,2], Marcus Liwicki[1]**
[1]University of Kaiserslautern, Germany
[2]German Research Center for Artificial Intelligence (DFKI), Germany.
{federico.raue,sebastian.palacio,andreas.dengel}@dfki.de,
liwicki@cs.uni-kl.de

## ABSTRACT

The goal of this paper is to train a model based on the relation between two instances that represent the same *unknown* class. This scenario is inspired by the *Symbol Grounding Problem* and the *association learning* in infants. We propose a novel model called *Classless Association*. It has two parallel Multilayer Perceptrons (MLP) that uses one network as a target of the other network, and vice versa. In addition, the presented model is trained based on an EM-approach, in which the output vectors are matched against a statistical distribution. We generate four *classless* datasets based on MNIST, where the input is two different instances of the same digit. In addition, the digits have a uniform distribution. Furthermore, our *classless association* model is evaluated against two scenarios: totally supervised and totally unsupervised. In the first scenario, our model reaches a good performance in terms of accuracy and the classless constraint. In the second scenario, our model reaches better results against two clustering algorithms.

## 1 INTRODUCTION

Infants are able to learn the binding between *abstract concepts* to the real world via their sensory input. For example, the abstract concept *ball* is binding to the visual representation of a rounded object and the auditory representation of the phonemes */b/ /a/ /l/*. This scenario can be seen as the *Symbol Grounding Problem* (Harnad, 1990). Moreover, infants are also able to learn the *association* between different sensory input signals while they are still learning the binding of the abstract concepts. Several results have shown a correlation between object recognition (visual) and vocabulary acquisition (auditory) in infants (Balaban & Waxman, 1997; Asano et al., 2015). One example of this correlation is the first words that infants have learned. In that case, the words are mainly nouns, which are *visible concepts*, such as, dad, mom, ball, dog, cat (Gershkoff-Stowe & Smith, 2004). As a result, we can define the previous scenario in terms of a machine learning tasks. More formally, the task is defined by learning the association between two parallel streams of data that represent the same *unknown* class (or abstract concept). Note that this task is different from the *supervised* association where the data has labels. First, the semantic concept is unknown in our scenario whereas it is known in the supervised case. Second, both classifiers needs to agree on the same coding scheme for each sample pair during training. In contrast, the coding-scheme is already pre-defined before training in the supervised case. Figure 1 shows an example of the difference between a supervised association task and our scenario.

Usually, classifiers requires labeled data for training. However, the presented scenario needs an alternative training mechanism. One way is to train based on statistical distributions. Casey (1986) proposed to solve the OCR problem using language statistics for inferring form images to characters. Later on, Knight et al. (2006) applied a similar idea to machine translation. Recently, Sutskever et al. (2015) defined the *Output Distribution Matching (ODM)* cost function for dual autoencoders and generative networks.

In this paper, we are proposing a novel model that is trained based on the association of two input samples of the same *unknown* class. The presented model has two parallel Multilayer Perceptrons (MLPs) with an Expectation-Maximization (EM) (Dempster et al., 1977) training rule that matches the network output against a statistical distribution. Also, both networks agree on the same classification because one network is used as target of the other network, and vice versa. Our model has some

| Task | Input Samples *(same)* | Abstract Concept Association *(different)* | Coding Scheme for each input *(different)* | Classifiers *(same)* |
|---|---|---|---|---|
| **Supervised Association** | 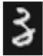 <br>  | "three" <br><br> "three" | Known *before* and *after* Training <br>  | classifier 1 <br><br> classifier 2 |
| **Classless Association (our work)** |  <br>  | **"unknown"** <br><br> **"unknown"** | Unknown *before* Training     Known *after* Training <br>  | classifier 1 <br><br> classifier 2 |

Figure 1: Difference between the *supervised* and *classless* association tasks. The classless association is more challenging that the supervised association because the model requires to learn to discriminate the semantic concept without labels. In addition, both classifiers need to agree on the same coding scheme for each semantic concept. In contrast, the mentioned information is already known in the supervised association scenario.

similarities with *Siamese Networks* proposed by Chopra et al. (2005). They introduced their model for supervised face verification where training is based on constraints of pairs of faces. The constraints exploit the relation of two faces that may or may not be instances of the same person. However, there are some differences to our work. First, our training rule does not have *pre-defined* classes before training, whereas the *Siamese Network* requires labeled samples. Second, our model only requires instances of the same unknown class, whereas the Siamese network requires two types of input pairs: a) instances of the same person and b) instances of two different persons. Our contributions in this paper are

- We define a novel training rule based on *matching* the output vectors of the presented model and a statistical distribution. Note that the output vectors are used as symbolic features similar to the *Symbol Grounding Problem*. Furthermore, the proposed training rule is based on an *EM-approach* and classified each sample based on generated *pseudo-classes* (Section 2.1).

- We propose a novel architecture for learning the association in the *classless* scenario. Moreover, the presented model uses two parallel MLPs, which require to agree on the same class for each input sample. This association is motivated by the correlation between different sensory input signals in infants development. In more detail, one network is the target of the other network, and vice versa. Also, note that our model is gradient-based and can be extended to deeper architectures (Section 2.2).

- We evaluate our *classless* association task against two cases: totally supervised and totally unsupervised. In this manner, we can verify the range of our results in terms of supervised and unsupervised cases since our model is neither totally supervised nor totally unsupervised. We compare against a MLP trained with labels as the supervised scenario (upper bound) and two clustering algorithms (K-means and Hierarchical Agglomerative) as the unsupervised scenario (lower bound). First, our model reaches better results than the clustering. Second, our model shows promising results with respect to the supervised scenario (Sections 3 and 4).

## 2 METHODOLOGY

In this paper, we are interested in the *classless* association task in the following scenario: two input instances $x^{(1)}$ and $x^{(2)}$ belong to the same unknown class $c$, where $x^{(1)} \in X^{(1)}$ and $x^{(2)} \in X^{(2)}$ are two disjoint sets, and the goal is to learn the output classification of $x^{(1)}$ and $x^{(2)}$ is the same $c^{(1)} = c^{(2)}$, where $c^{(1)}$ and $c^{(2)} \in C$ is the set of possible classes. With this in mind, we present a model that has two parallel *Multilayer Perceptrons (MLPs)* that are trained with an *EM-approach* that associates both networks in the following manner: one network uses the other network as a target, and vice versa. We explain how the output vectors of the network are matched to a statistical distribution in Section 2.1 and the classless association learning is presented in Section 2.2.

### 2.1 STATISTICAL CONSTRAINT

One of our constraint is to train a MLP without classes. As a result, we use an alternative training rule based on matching the output vectors and a statistical distribution. For simplicity, we explain our training rule using a single MLP with one hidden layer, which is defined by

$$z = network(x; \theta) \tag{1}$$

where $x \in \mathbb{R}^n$ is the input vector, $\theta$ encodes the parameters of the MLP, and $z \in \mathbb{R}^c$ is the output vector. Moreover, the output vectors $(z_1, \ldots, z_m)$ of a mini-batch of size $m$ are matched to a target distribution $(\mathbb{E}[z_1, \ldots, z_m] \sim \phi \in \mathbb{R}^c)$, e.g., uniform distribution. We have selected a uniform distribution because it is an ideal case to have a balanced dataset for any classifier. However, it is possible to extend to different distribution. We introduce a new parameter that is a *weighting vector* $\gamma \in \mathbb{R}^c$. The intuition behind it is to guide the network based on a set of generated *pseudo-classes* $c$. These *pseudo-classes* can be seen as cluster indexes that group similar elements. With this in mind, we also propose an EM-training rule for learning the unknown class given a desired target distribution. We want to point out that the *pseudo-classes* are internal representations of the network that are independent of the labels.

The *E-step* obtains the current statistical distribution given the output vectors $(z_1, \ldots, z_m)$ and the *weighting vector* $(\gamma)$. In this case, an approximation of the distribution is obtained by the following equation

$$\hat{z} = \frac{1}{M} \sum_{i=1}^{M} power(z_i, \gamma) \tag{2}$$

where $\gamma$ is the weighting vector, $z_i$ is the output vector of the network, $M$ is the number of elements, and the function $power$[1] is the element-wise power operation between the output vector and the weighting vector. We have used the *power function* because the output vectors $(z_1, \ldots, z_m)$ are quite similar between them at the initial state of the network, and the *power function* provides an initial boost for learning to separate the input samples in different *pseudo-classes* in the first iterations. Moreover, we can retrieve the *pseudo-classes* by the maximum value of the following equation

$$c^* = arg\,max_c\,power(z_i, \gamma) \tag{3}$$

where $c^*$ is the *pseudo-class*, which are used in the *M-step* for updating the MLP weights. Also, note that the *pseudo-classes* are not updated in an online manner. Instead, the *pseudo-classes* are updated after a certain number of iterations. The reason is the network requires a number of iterations to learn the common features.

The *M-step* updates the weighting vector $\gamma$ given the current distribution $\hat{z}$. Also, the MLP parameters $(\theta)$ are updated given the current classification given by the *pseudo-classes*. The cost function is the variance between the distribution and the desired statistical distribution, which is defined by

$$cost = (\hat{z} - \phi)^2 \tag{4}$$

---

[1]We decide to use *power* function instead of $z_i^\gamma$ in order to simplify the index notation

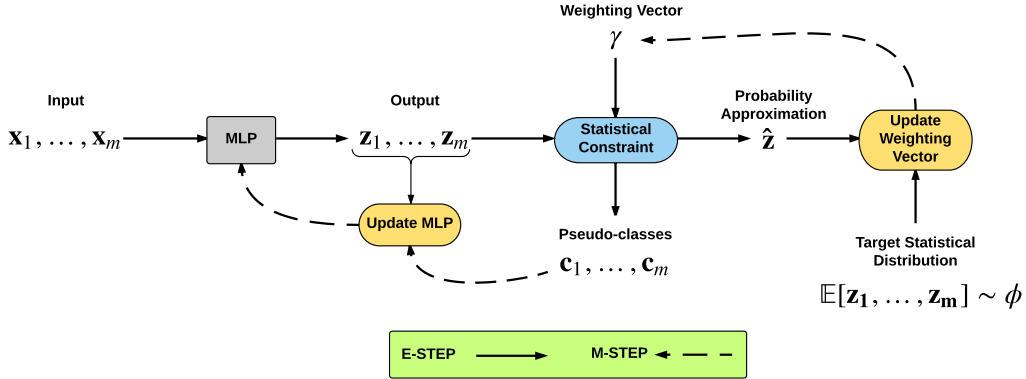

Figure 2: The proposed training rule applied to a single MLP. E-steps generates a set of pseudo-classes $c_1, \ldots, c_m$ for each output in the mini-batch of size $m$, and a probability approximation $\hat{z}$ of the output vectors in the mini-batch. M-step updates the MLP weights given the pseudo-classes and the weighting vector $\gamma$ giving the target statistical distribution $\phi$.

where $\hat{z}$ is the current statistical distribution of the output vectors, and $\phi$ is a vector that represent the desired statistical distribution, e.g. uniform distribution. Then, the weighting vector is updated via gradient descent

$$\gamma = \gamma - \alpha \, * \nabla_\gamma cost \tag{5}$$

where $\alpha$ is the learning rate and $\nabla_\gamma cost$ is the derivative w.r.t $\gamma$. Also, the MLP weights are updated via the generated *pseudo-classes*, which are used as targets in the backpropagation step.

In summary, we propose an EM-training rule for matching the network output vectors and a desired target statistical distribution. The *E-Step* generates *pseudo-classes* and finds an approximation of the current statistical distribution of the output vectors. The *M-Step* updates the MLP parameters and the weighting vector. With this in mind, we adapt the mentioned training rule for the classless association task. Figure 2 summarizes the presented EM training rule and its components.

## 2.2 CLASSLESS ASSOCIATION LEARNING

Our second constraint is to classify both input samples as the same class and different from the other classes. Note that the *pseudo-class* (Equation 3) is used as identification for each input sample and it is not related to the semantic concept or labels. The presented classless association model is trained based on a statistical constraint. Formally, the input is represented by the pair $x^{(1)} \in \mathbb{R}^{n1}$ and $x^{(2)} \in \mathbb{R}^{n2}$ where $x^{(1)}$ and $x^{(2)}$ are two different instances of the same *unknown* label. The classless association model has two parallel Multilayer Perceptron $MLP^{(1)}$ and $MLP^{(2)}$ with training rule that follows an EM-approach (*cf.* Section 2.1). Moreover, input samples are divided into several mini-batches of size $m$.

Initially, all input samples have random *pseudo-classes* $c_i^{(1)}$ and $c_i^{(2)}$. The pseudo-classes have the same desired statistical distribution $\phi$. Also, the weighting vectors $\gamma^{(1)}$ and $\gamma^{(2)}$ are initialized to one. Then each input element from the mini-batch is propagated forward to each MLP. Afterwards, an estimation of the statistical distribution for each MLP ($\hat{z}^{(1)}$ and $\hat{z}^{(2)}$) is obtained. Furthermore, a new set of *pseudo-classes* ($c_1^{(1)}, \ldots, c_m^{(1)}$ and $c_1^{(2)}, \ldots, c_m^{(2)}$) is obtained for each network. Note that this first part can be seen as an *E-step* from Section 2.1. We want to point out that the pseudo-classes are updated only after a number of iterations.

The second part of our association training updates the MLP parameters and the *weighting vector* ($\gamma^{(1)}$ and $\gamma^{(2)}$). In this step, one network ($MLP^{(1)}$) uses pseudo-classes ($c_1^{(2)}, \ldots, c_m^{(2)}$) obtained from the other network ($MLP^{(2)}$), and vice versa. In addition, the weighting vector is updated

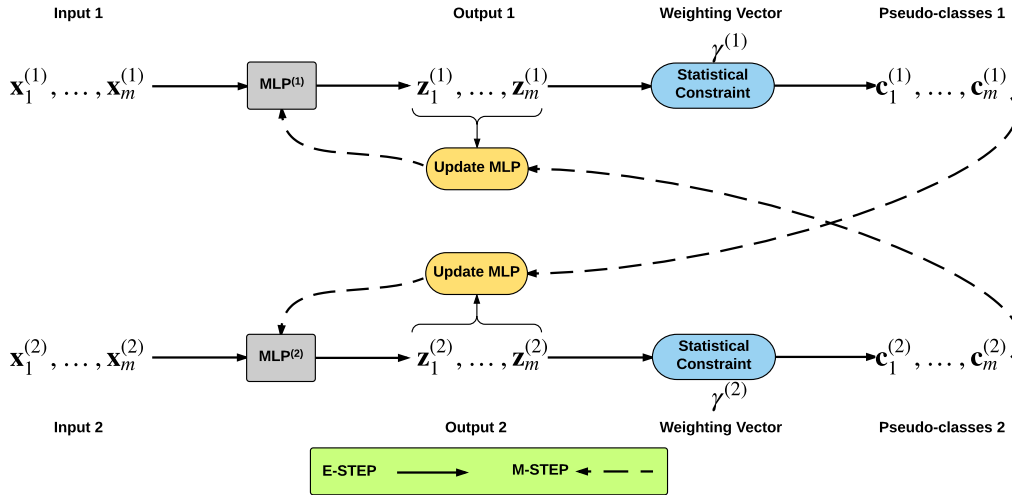

Figure 3: Overview of the presented model for classless association of two input samples that represent the same unknown classes. The association relies on matching the network output and a statistical distribution. Also, it can be observed that our model uses the pseudo-classes obtained by $MLP^{(1)}$ as targets of $MLP^{(2)}$, and vice versa.

between the output approximation ($\hat{z}^{(1)}$ and $\hat{z}^{(2)}$) and the desired target distribution ($\phi$). Figure 3 shows an overview of the presented model.

## 3  EXPERIMENTS

In this paper, we are interested in a simplified scenario inspired by the *Symbol Grounding Problem* and the association learning between sensory input signal in infants. We evaluated our model in four *classless* datasets that are generated from MNIST (Lecun & Cortes, 2010). The procedure of generating *classless datasets* from *labeled datasets* have been already applied in (Sutskever et al., 2015; Hsu & Kira, 2015). Each dataset has two disjoint sets *input 1* and *input 2*. The first dataset (*MNIST*) has two different instances of the same digit. The second dataset (*Rotated-90 MNIST*) has two different instances of the same digit, and all input samples in *input 2* are rotated 90 degrees. The third dataset (*Inverted MNIST* ) follows a similar procedures as the second dataset, but the transformation of the elements in *input 2* is the invert function instead of rotation. The last dataset (*Random Rotated MNIST*) is more challenging because all elements in *input 2* are randomly rotated between 0 and $2\pi$. All datasets have a uniform distribution between the digits and the dataset size is 21,000 samples for training and 4,000 samples for validation and testing.

The following parameters turned out being optimal on the validation set. For the first three datasets, each internal MLP relies on two fully connected layers of 200 and 100 neurons respectively. The learning rate for the MLPs was set to start at 1.0 and was continuously decaying by half after every 1,000 iterations. We set the initial *weighting vector* to 1.0 and updated after every 1,000 iterations as well. Moreover, the best parameters for the fourth dataset were the same for $MLP^{(1)}$ and different for $MLP^{(2)}$, which has two fully connected layers of 400 and 150 neurons respectively and the learning rate stars at 1.2. The target distribution $\phi$ is uniform for all datasets. The decay of the learning rate (Equation 5) for the *weighting vector* was given by $1/(100+epoch)^{0.3}$, where $epoch$ was the number of training iterations so far. The mini-batch size $M$ is 5,250 sample pairs (corresponding to 25% of the training set) and the mean of the derivatives for each mini-batch is used for the back-propagation step of MLPs. Note that the mini-batch is quite big comparing with common setups. We infer from this parameter that the model requires a sample size big enough for estimating the uniform distribution and also needs to learn slower than traditional approaches. Our model was implemented in *Torch*.

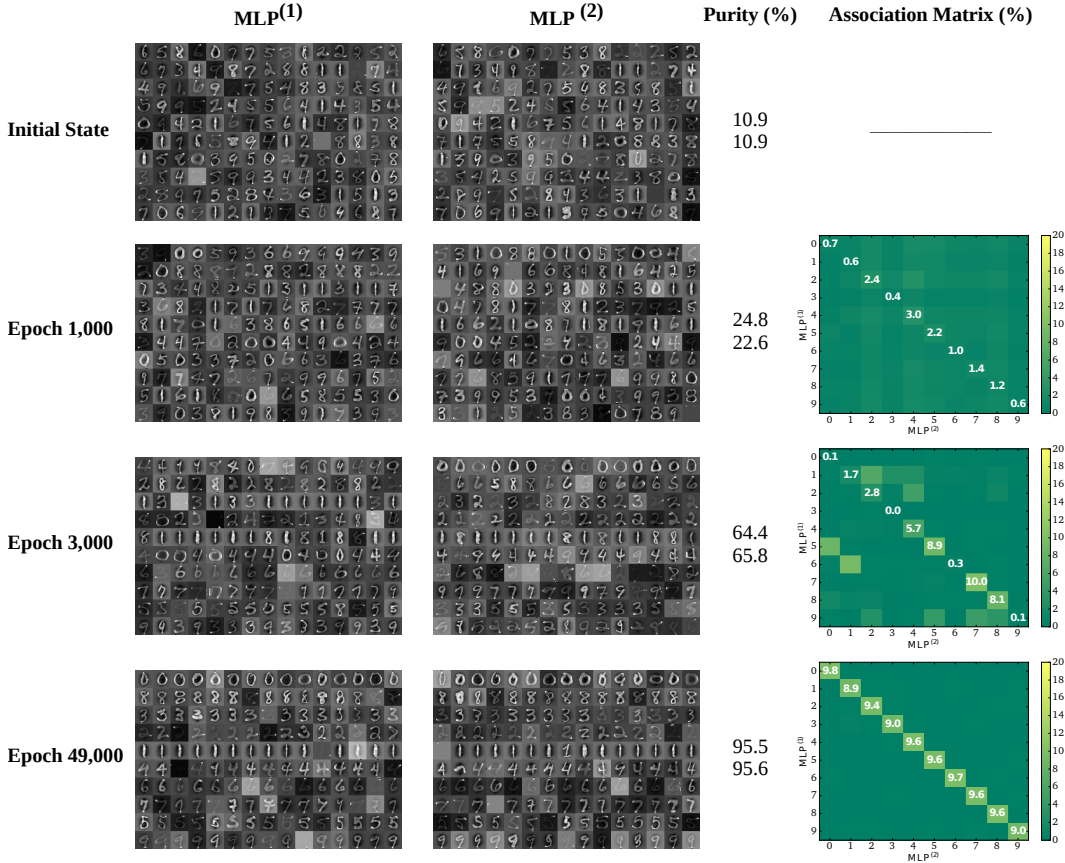

Figure 4: Example of the presented model during *classless* training. In this example, there are ten *pseudo-classes* represented by each row of $MLP^{(1)}$ and $MLP^{(2)}$. Note that the output classification are randomly selected (not cherry picking). Initially, the *pseudo-classes* are assigned randomly to all input pair samples, which holds a uniform distribution (first row). Then, the *classless association* model slowly start learning the features and grouping similar input samples. Afterwards, the output classification of both MLPs slowly agrees during training, and the association matrix shows the relation between the occurrences of the *pseudo-classes*.

To determine the baseline of our *classless constraint*, we compared our model against two cases: totally supervised and totally unsupervised. In the supervised case, we used the same MLP parameters and training for a fair comparison. In the unsupervised scenario, we used k-means and agglomerative clustering to each set (*input 1* and *input 2*) independently. The clustering algorithm implementation are provided by scikit-learn (Pedregosa et al., 2011).

## 4 RESULTS AND DISCUSSION

In this work, we have generated ten different folds for each dataset and report the average results. We introduce the *Association Accuracy* for measuring association, and it is defined by the following equation

$$Association\ Accuracy = \frac{1}{N} \sum_{i=1}^{N} \mathbb{1}(c_i^{(1)} = c_i^{(2)}) \qquad (6)$$

Table 1: Association Accuracy (%) and Purity (%) results. Our model is compared with the supervised scenario (class labels are provided) and with K-means and Hierarchical Agglomerative clustering (no class information).

| Dataset | Model | Association Accuracy (%) | Purity (%) | |
|---|---|---|---|---|
| | | | input 1 | input 2 |
| MNIST | supervised association | $96.7 \pm 0.3$ | $96.7 \pm 0.2$ | $96.6 \pm 0.3$ |
| | **classless association** | $87.4 \pm 2.9$ | $87.1 \pm 6.6$ | $87.0 \pm 6.4$ |
| | K-means | - | $63.9 \pm 2.2$ | $62.5 \pm 3.7$ |
| | Hierarchical Agglomerative | - | $64.9 \pm 4.7$ | $64.3 \pm 5.5$ |
| Rotated-90 MNIST | supervised association | $93.2 \pm 0.3$ | $96.4 \pm 0.2$ | $96.6 \pm 0.21$ |
| | **classless association** | $86.5 \pm 2.5$ | $82.9 \pm 4.5$ | $82.9 \pm 4.3$ |
| | K-means | - | $65.0 \pm 2.8$ | $64.0 \pm 3.6$ |
| | Hierarchical Agglomerative | - | $65.4 \pm 3.5$ | $64.1 \pm 4.1$ |
| Inverted MNIST | supervised association | $93.2 \pm 0.3$ | $96.5 \pm 0.2$ | $96.5 \pm 0.2$ |
| | **classless association** | $89.2 \pm 2.4$ | $89.0 \pm 6.8$ | $89.1 \pm 6.8$ |
| | K-means | - | $64.8 \pm 2.0$ | $65.0 \pm 2.5$ |
| | Hierarchical Agglomerative | - | $64.8 \pm 4.4$ | $64.4 \pm 3.8$ |
| Random Rotated MNIST | supervised association | $88.0 \pm 0.5$ | $96.5 \pm 0.3$ | $90.9 \pm 0.5$ |
| | **classless association** | $69.3 \pm 2.2$ | $75.8 \pm 7.3$ | $65.3 \pm 5.0$ |
| | K-means | - | $64.8 \pm 2.6$ | $14.8 \pm 0.4$ |
| | Hierarchical Agglomerative | - | $65.9 \pm 2.8$ | $15.2 \pm 0.5$ |

where the *indicator function* is one if $c_i^{(1)} = c_i^{(2)}$, zero otherwise; $c_i^{(1)}$ and $c_i^{(2)}$ are the *pseudo-classes* for $MLP^{(1)}$ and $MLP^{(2)}$, respectively, and $N$ is the number of elements. In addition, we also reported the *Purity* of each set (*input* 1 and *input* 2). *Purity* is defined by

$$Purity(\Omega, \mathcal{C}) = \frac{1}{N} \sum_{i=1}^{k} max_j |c_i \cap gt_j| \qquad (7)$$

where $\Omega = \{gt_1, gt_2, \ldots, gt_j\}$ is the set of ground-truth labels and $\mathcal{C} = \{c_1, c_2, \ldots, c_k\}$ is the set of *pseudo-classes* in our model or the set of cluster indexes of K-means or Hierarchical Agglomerative clustering, and $N$ is the number of elements.

Table 1 shows the *Association Accuracy* between our model and the supervised association task and the *Purity* between our model and two clustering algorithms. First, the supervised association task performances better that the presented model. This was expected because our task is more complex in relation to the supervised scenario. However, we can infer from our results that the presented model has a good performance in terms of the classless scenario and supervised method. Second, our model not only learns the association between input samples but also finds similar elements covered under the same *pseudo-class*. Also, we evaluate the purity of our model and found that the performance of our model reaches better results than both clustering methods for each set (*input* 1 and *input* 2).

Figure 4 illustrates an example of the proposed learning rule. The first two columns ($MLP^{(1)}$ and $MLP^{(2)}$) are the output classification (Equation 3) and each row represents a *pseudo-class*. We have randomly selected 15 output samples for each MLP (not cherry picking). Initially, the *pseudo classes* are random selected for each MLP. As a result, the output classification of both networks does not show any visible discriminant element and the initial purity is close to random choice (first row). After 1,000 epochs, the networks start learning some features in order to discriminate the input samples. Some groups of digits are grouped together after 3,000 epochs. For example, the first row of $MLP^{(2)}$ shows several digits *zero*, whereas $MLP^{(1)}$ has not yet agree on the same digit for that *pseudo-class*. In contrast, both MLPs have almost agree on digit *one* at the fifth row. Finally, the association is learned using only the statistical distribution of the input samples and each digit is represented by each *pseudo-class*.

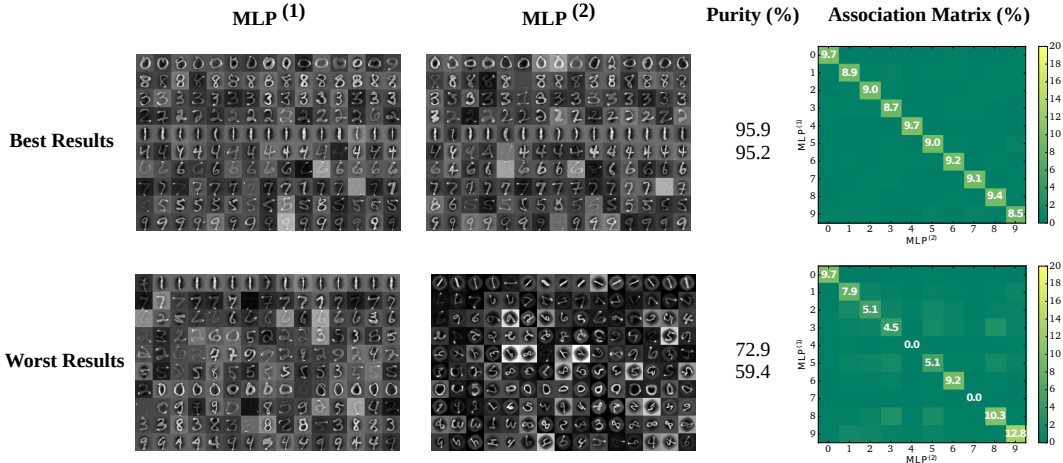

Figure 5: Example of the best and worst results among all folds and datasets. It can be observed our model is able to learn to discriminate each digit (first row). However, the presented model has a limitation that two or more digits are assigned to the same *pseudo-class* (last row of $MLP^{(1)}$ and $MLP^{(2)}$).

Figure 5 shows the best and worst results of our model in two cases. The first row is the best result from *MNIST dataset*. Each row of $MLP^{(1)}$ and $MLP^{(2)}$ represent a *pseudo-class*, and it can be observed that all digits are grouped together. In addition, the association matrix shows a distribution per digit close to the desired uniform distribution, and the purity of each input is close to the supervised scenario. In contrast, the second row is our worst result from *Random Rotated MNIST dataset*. In this example, we can observe that some digits are recognized by the same *pseudo-class*, for example, digit *one* and *seven* (first two rows). However, there two or more digits that are recognized by the same *pseudo-class*. For example, the last row shows that *nine* and *four* are merged. Our model is still able to reach better results than the unsupervised scenario.

## 5 CONCLUSION

In this paper, we have shown the feasibility to train a model that has two parallel MLPs under the following scenario: pairs of input samples that represent the same unknown classes. This scenario was motivated by the *Symbol Grounding Problem* and association learning between sensory input signal in infants development. We proposed a model based on gradients for solving the *classless association*. Our model has an EM-training that matches the network output against a statistical distribution and uses one network as a target of the other network, and vice versa. Our model reaches better performance than K-means and Hierarchical Agglomerative clustering. In addition, we compare the presented model against a supervised method. We find that the presented model with respect to the supervised method reaches good results because of two extra conditions in the unsupervised association: unlabeled data and agree on the same *pseudo-class*. We want to point out that our model was evaluated in an optimal case where the input samples are uniform distributed and the number of classes is known. However, we will explore the performance of our model if the number of classes and the statistical distrubtion are unknown. One way is to change the number of *pseudo-classes*. This can be seen as changing the number of clusters $k$ in k-means. With this in mind, we are planning to do more exhaustive analysis of the learning behavior with deeper architectures. Moreover, we will work on how a small set of labeled classes affects the performance of our model (similar to semi-supervised learning). Furthermore, we are interested in replicating our findings in more complex scenarios, such as, multimodal datasets like TVGraz (Khan et al., 2009) or Wikipedia featured articles (Rasiwasia et al., 2010). Finally, our work can be applied to more *classless* scenarios where the data can be extracted simultaneously from different input sources at the same time. Also, transformation functions can be applied to input samples for creating the association without classes.

ACKNOWLEDGMENTS

We would like to thank Damian Borth, Christian Schulze, Jörn Hees, Tushar Karayil, and Philipp Blandfort for helpful discussions.

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

## SUPPLEMENTAL MATERIAL

We have included two more examples of the *classless* training. In addition, we have generated some demos that show the training algorithm (`https://goo.gl/xsmkFD`)

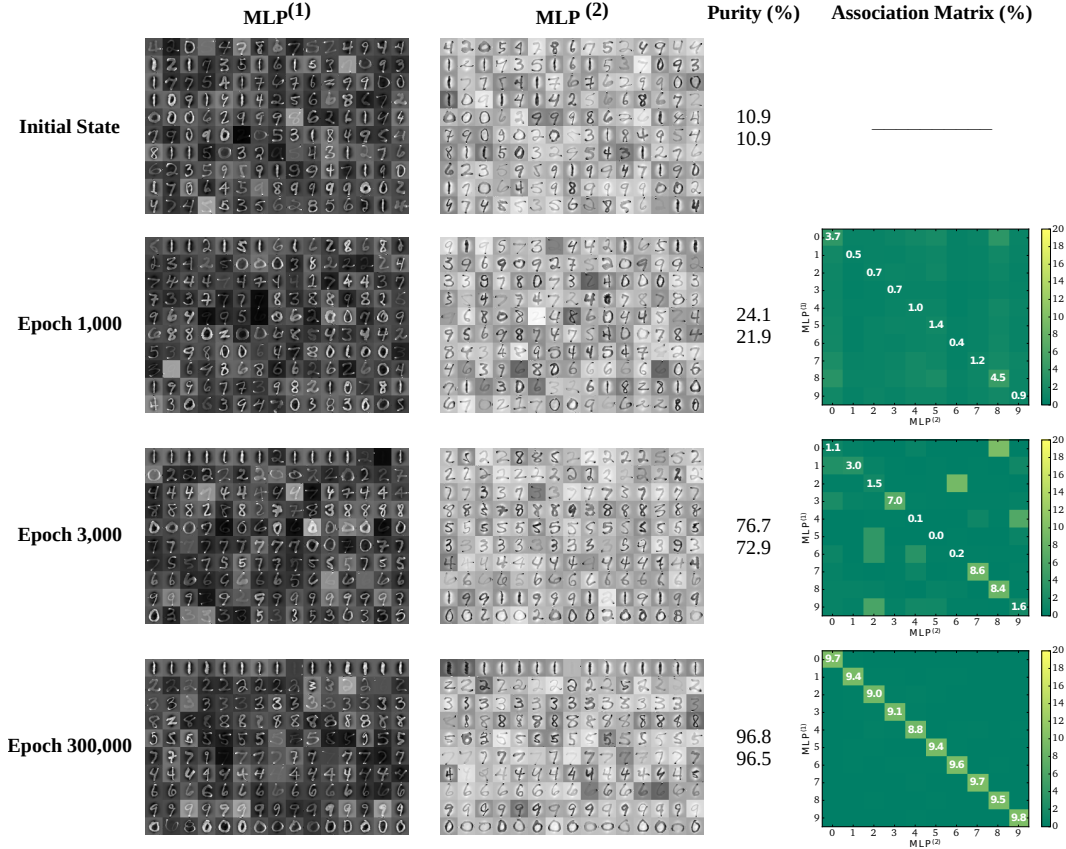

Figure 1: Example of the *classless* training using Inverted MNIST dataset.

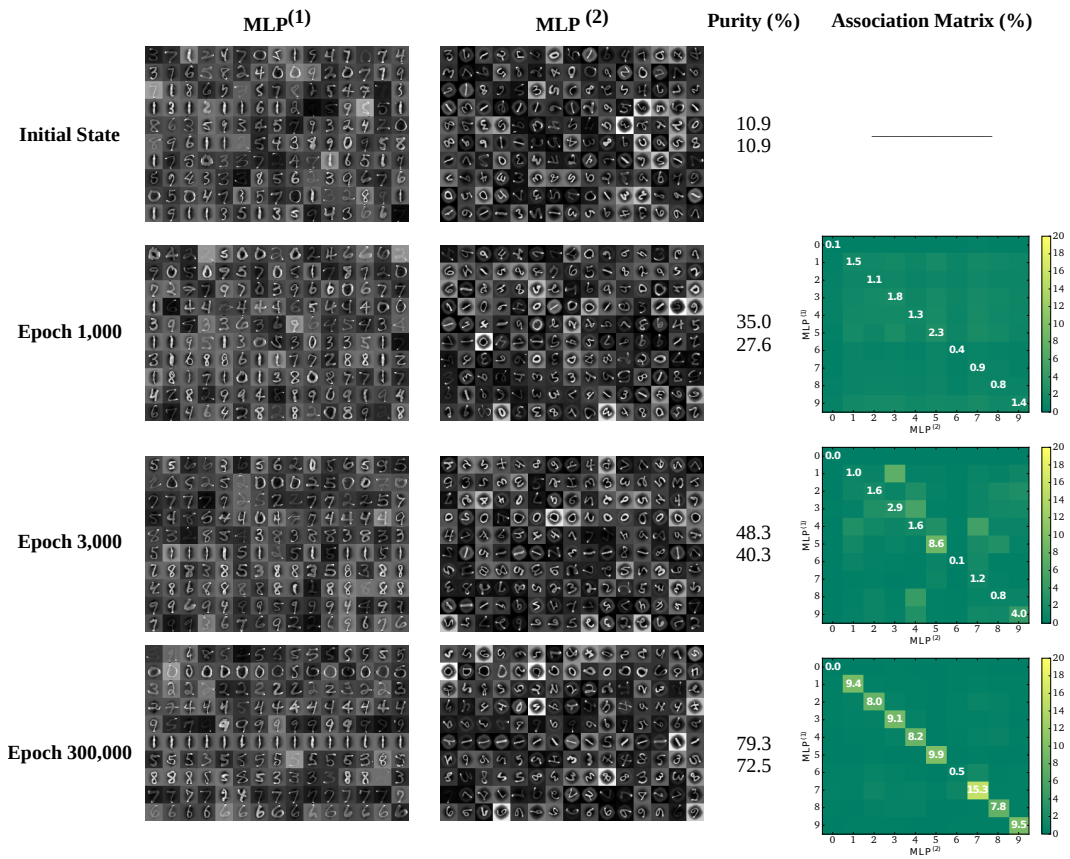

Figure 2: Example of the *classless* training using Random Rotated MNIST dataset.

