# Peer review of "Classless Association using Neural Networks"

_ICLR 2017 — rejected_

[Official Review · AnonReviewer2 · rating 6 · confidence 3 · 16 Dec 2016]
**Classes association**

The paper looks correct but still i am not convinced about the experimentation performed. Perhaps another experiment with more challenging data would be welcome. Honestly i don't find a clear motivation for this work however it could have some potential and it would be interested to be presented in conference.

[Official Review · AnonReviewer3 · rating 5 · confidence 3 · 17 Dec 2016]
**No Title**

The paper explores a new technique for classless association, a milder unsupervised learning where we do not know the class labels exactly, but we have a prior about the examples that belong to the same class. Authors proposed a two stream architecture with two neural networks, as streams process examples from the same class simultaneously. Both streams rely on the target (pseudo classes or cluster indices) of each other, and the outputs an intermediate representation z, which is forced to match with a statistical distribution (uniform in their case). The model is trained with EM where the E step obtains the current statistical distribution given output vectors z, and M step updates the weights of the architecture given z and pseudo-classes. Experimental results on re-organized MNIST exhibits better performance compared to classical clustering algorithms (in terms of association accuracy and purity). The authors further provide comparison against a supervised method, where proposed architecture expectedly performs worse but with promising results.

The basic motivation of the architecture apparently relies on unlabeled data and agreement of the same pseudo-labels generated by two streams. But the paper is hard to follow and the motivation for the proposed architecture itself, is hidden in details. What is trying to be achieved by matching distributions and using the pseudo-targets of the each other? Perhaps the statistical distribution of the classes is assumed to be uniform but how will it extend to other priors, or even the case where we do not assume that we know the prior? The current setup needs justifications. 

What would be very interesting is to see two examples having the same class but one from MNIST, the other from Rotated-MNIST or Background-MNIST. Because it is hard to guess how different the examples in two streams. 

At the end, I feel like the authors have found a very interesting approach for classless association which can be extended to lots of many-to-one problems. This is a good catch. I would like to see the idea in the future with some extensive experiments on large scale datasets and tasks. But the current version lacks the theoretical motivations and convincing experiments. I would definitely recommend this paper to be presented in ICLR workshop.

Few more points:
Typo: Figure1. second line in the caption "that" -> "than"
Necessity of Equation 2 is not clear
Batch size M is enormous compared to classical models, there is no explanation for this
Why uniform? should be clarified (of course it is the simplest prior to pick but just a few words about it would be good for completeness)
Typo: Page 6, second paragraph line 3: "that" -> "than"

[Official Review · AnonReviewer1 · rating 5 · confidence 4 · 20 Dec 2016]

The paper presents an alternative way of supervising the training of neural network without explicitly using labels when only link/not-link information is available between pairs of examples. A pair of network is trained each of which is used to supervise the other one.


The presentation of the paper is not very clear, the writing can be improved.
Some design choice are not explained: Why is the power function used in the E-step for approximating the distribution (section 2.1)? Why do the authors only consider a uniform distribution? I understand that using a different prior breaks the assumption that nothing is known about the classes. However I do not see a practical situations where the proposed setting/work would be useful.  

Also, there exist a large body of work in semi-supervised learning with co-training based on a similar idea. 

Overall, I think this work should be clarified and improved to be a good fit for this venue.

[Author Response · Federico Raue · 13 Jan 2017]
**New revision**

We have updated our paper.  The changes are
* We have improved the clarity and motivation of our model
* We have evaluated our model to three more classless datasets (Rotated-90 MNIST, Inverted MNIST, and Random Rotation MNIST).
* We have updated Figure 4 and 5 for showing some random output classification samples instead of the mean of all images.
* We have added two more examples and demo as supplemental material

[Final Decision · Program Chairs · 06 Feb 2017]
**ICLR committee final decision**

The paper explores neural-network learning on pairs of samples that are labeled as either similar or dissimilar. The proposed model appears to be different from standard siamese architectures, but it is poorly motivated. The experimental evaluation of the proposed model is very limited.